# Hypovitaminosis D and Cardiometabolic Risk Factors in Adolescents with Severe Obesity

**DOI:** 10.3390/children7020010

**Published:** 2020-02-01

**Authors:** Teodoro Durá-Travé, Fidel Gallinas-Victoriano, Diego Mauricio Peñafiel-Freire, María Urretavizcaya-Martinez, Paula Moreno-González, María Jesús Chueca-Guindulain

**Affiliations:** 1Department of Pediatrics, School of Medicine, University of Navarra, 31008 Pamplona, Spain; 2Department of Pediatrics, Navarra Hospital Complex, 31008 Pamplona, Spain; fivictoriano@hotmail.com (F.G.-V.); Dpfreire.89@gmail.com (D.M.P.-F.); mariaurretavizcayamartinez@gmail.com (M.U.-M.); paulamoreno_91@hotmail.com (P.M.-G.); mj.chueca.guindulain@cfnavarra.es (M.J.C.-G.); 3Navarra Institute for Health Research (IdisNA), 31008 Pamplona, Spain

**Keywords:** adolescents, severe obesity, cardiometabolic risks factors, Vitamin D

## Abstract

Background/Objectives. Obesity is associated with cardiometabolic risk factors and with Vitamin D deficiency. The aim of this study was to examine the relationship between 25(OH)D concentrations and cardiometabolic risk factors in adolescents with severe obesity. Subjects/Methods. A cross-sectional clinical assessment (body mass index, fat mass index, fat-free mass index, waist-to-height ratio, and blood pressure) and metabolic study (triglycerides, total cholesterol, HDL-C, LDL-C, glucose, insulin, HOMA-IR, leptin, calcium, phosphorous, calcidiol, and PTH) were carried out in 236 adolescents diagnosed with severe obesity (BMI z-score > 3.0, 99th percentile), aged 10.2–15.8 years. The criteria of the US Endocrine Society were used for the definition of Vitamin D status. Results. Subjects with Vitamin D deficiency had significantly elevated values (*p* < 0.05) for BMI z-score, waist circumference, waist z-score, body fat percentage, fat mass index, systolic and diastolic blood pressure, total cholesterol, triglycerides, LDL-C, insulin, HOMA-IR, leptin, and PTH than subjects with normal Vitamin D status. There was a significant negative correlation (*p* < 0.05) of serum 25(OH)D levels with body fat percentage, FMI, systolic BP, total cholesterol, triglyceride, LDL-C, glucose, insulin, HOMA-IR, leptin, and PTH. Conclusions. Low Vitamin D levels in adolescents with severe obesity were significantly associated with some cardiometabolic risk factors, including body mass index, waist circumference, fat mass index, high blood pressure, impaired lipid profile, and insulin resistance.

## 1. Introduction

Severe childhood obesity is associated with an increased prevalence of immediate risk of cardiometabolic complications [1,2]. Longitudinal studies have shown the trend of developing severe obesity in adulthood when severe obesity is present in childhood [3,4]. In the same way, tracking studies have shown that this early acquisition of risk factors persists in adulthood, including elevated levels of blood pressure, increased lipid serum concentration, insulin resistance, and manifestations of metabolic syndrome [1,4,5] Therefore, modifiable cardiometabolic risk factors should be promptly identified in children with severe obesity.

On the other hand, obesity is a known risk factor for Vitamin D deficiency. In fact, circulating concentration of 25(OH)D is inversely associated with the severity of obesity [6,7,8]. Data on the relationship between hypovitaminosis D and the different components of metabolic syndrome, both in adults [9,10,11] and children with obesity [12,13,14,15,16] are, at present, inconclusive.

Authors do not generally distinguish between obesity and severe obesity, and this detail could be of practical interest. In fact, children with severe obesity constitute a subgroup with the highest risk of cardiometabolic disease (dyslipidemia, insulin resistance, arterial hypertension, etc.) and Vitamin D deficiency, and they would be the optimum population group in order to attempt to analyze a relationship between both entities.

The objective of the study was to examine the relationship between 25(OH)D concentrations and cardiometabolic risk factors in adolescents with severe obesity.

## 2. Material and Methods

### 2.1. Patients

This is a cross-sectional study carried out in a sample of 236 adolescents (154 boys and 82 girls) diagnosed with severe obesity (BMI z-score > 3.0, 99th percentile), aged 10.2–15.8 years. All patients involved in the study were Caucasian and showed pubertal changes (Tanner stages: II–V); and they passed a clinical examination and blood testing. The attention was provided in the Pediatric Endocrinology Unit of the Navarra Hospital Complex (Pamplona, Spain), in the period January 2015–December 2018.

Participants had no record of any illness affecting bone health or chronic pathologies that might alter growth, body composition, food ingestion, or physical activity, nor had they received any medication (antiepileptic drugs or glucocorticoids), Vitamin D, or calcium supplements.

### 2.2. Clinical Assessment

The anthropometric measurements were taken according to a protocol that was previously published [17]. Information collected from every individual included weight, height, body mass index (BMI), skinfold thickness (biceps, triceps, subscapular, and suprailiac), and waist circumference. Weight and height measurements were taken in underwear and barefoot. An Año-Sayol scale was used for weight measurement (reading interval 0–120 kg and a precision of 100 g), and a Holtain wall stadiometer for height measurement (reading interval 60–210 cm and precision 0.1 cm). Subsequent calculations allowed evaluation of BMI by means of the following formula: weight (kg)/height^2^ (m).

Skinfold thickness values were measured with an accuracy of 0.1 mm on the left side of the body, using Holtain skinfold calipers (CMS Weighing Equipment, Crymych, United Kingdom). The body fat percentage (%), fat mass (kg), and fat-free mass (kg) were calculated by using the equations reported by Siri et al., adjusted for sex and age [18]. In the same way, the fat mass index (FMI) and the fat-free mass index (FFMI) were calculated by using the following formulas: fat mass (kg)/height^2^ (m) and free fat mass (kg)/height^2^ (m), respectively.

Waist circumference (WC) was registered by using a tape measure placed on a horizontal line equidistant from the last rib and the iliac crest, and the waist-to-height ratio (WtHR) was calculated according to the following formula: waist (m)/height^2^ (m). Measurements were performed by the same trained individual.

The Spanish Society of Pediatric Gastroenterology, Hepatology and Nutrition (Sociedad Española de Gastroenterología, Hepatología y Nutrición Pediátrica, available at http://www.gastroinf.es/nutritional/) provided the program Aplicación Nutricional for the assessment of z-score values for the BMI, skinfold thickness, and waist circumference. The graphics from Ferrández et al. (Centro Andrea Prader, Zaragoza 2002) were used as reference charts [19]. Severe obesity was defined by a BMI z-score higher than 3.0 (99th percentile).

Blood pressure (BP) was measured in the right arm, with the patient in the supine position, using a Visomat comfort 20/40 (Roche Diagnostics Inc., Amman, Jordan) digital blood pressure monitor, recording the lowest of three measurements. Arterial hypertension (HTA) was defined when systolic (SBP) and/or diastolic pressure (DBP) was equal to or higher than the 95th percentile by age, sex, and height, according to the American reference charts (National high blood pressure Program in Children and Adolescents) [20]. In summary, systolic blood pressure over 130 mm Hg or diastolic blood pressure over 85 mm Hg were the cutoff values for the consideration of arterial hypertension.

### 2.3. Metabolic Study

Blood testing allowed the determinations of plasma concentrations for glucose, insulin, triglycerides, total cholesterol, high-density lipoprotein cholesterol (HDL-C), low-density lipoprotein cholesterol (LDL-C), leptin, calcium, and phosphorus, which were measured under basal fasting conditions, using standardized methodologies. The determination of 25(OH)D levels was made by means of a high-specific chemiluminescence immunoassay (LIAISON Assay, Diasorin, Dietzenbach, Germany), and the determination of parathyroid hormone (PTH) levels required a highly specific solid-phase, two-site chemiluminescent enzyme-labeled immunometric assay in an IMMULITE analyzer (DPC Biermann, Bad Nauheim, Germany).

In order to evaluate insulin resistance, the HOMA-IR (Homeostasis Model Assessment of Insulin Resistance) indexes were calculated from fasting glucose and insulin concentrations (glucose levels in mmol × insulin in μUmL/22.5). An HOMA-IR value equal to or higher than 2.5 was considered to be insulin resistance [21].

According to the International Diabetes Federation consensus report for children and adolescents [22], serum TC levels higher than 200 mg/dL, TG levels higher than 150 mg/dL, LDL-C levels higher than 130 mg/dL, or HDL-C levels lower than 40 mg/dL were accepted as dyslipidemia, and fasting blood glucose higher than 100 mg/dL as dysglycaemia.

The distribution of individuals according to Vitamin D plasma levels followed the criteria of the United States Endocrine Society [23,24]. Specifically, 25(OH)D plasma levels lower than 20 ng/mL (<50 nmol/L) corresponded to Vitamin D deficiency, calcidiol levels between 20 and 29 ng/mL (50–75 nmol/L) to Vitamin D insufficiency, and concentrations equal to or higher than 30 ng/mL (>75 nmol/L) to Vitamin D sufficiency.

### 2.4. Statistical Analysis

Results are represented as percentages (%) and means (M), with their corresponding standard deviations (SD). The posterior statistical analysis (descriptive statistics, Student’s *t*-test, analysis of variance, χ^2^ test, and Pearson’s correlation) was done by using the software package *Statistical* Packages for the Social Sciences version 20.0 (Chicago, IL, USA). A probability value (*p*-value) of <0.05 was settled as the level of statistical significance.

Adequate information regarding proceedings and potential implications was given to the parents and/or legal guardians, and the corresponding consent was a requirement prior to the incorporation to this study in all cases. The study was submitted and approved of after the assessment of the Ethics Committee for Human Investigation of Navarra Hospital Complex (code: 14/209), in accordance with the ethical standards stated in the Declaration of Helsinki, 1964 and later amendments).

## 3. Results

Ninety-four (39.8%) individuals had Vitamin D deficiency, 92 (39%) Vitamin D insufficiency, and 50 (21.2%) Vitamin D sufficiency. The prevalence of hypovitaminosis in adolescents with severe obesity was 78.8%.

Table 1 shows and compares the mean values for anthropometric and biochemical parameters registered according to Vitamin D status. Subjects with Vitamin D deficiency have significantly high values (*p* < 0.05) for BMI z-score, waist circumference, waist z-score, body fat percentage, FMI, systolic and diastolic BP, total cholesterol, triglycerides, LDL-C, insulin, HOMA-IR, leptin, and PTH. There were no significant differences in mean values of age, weight z-score, height z-score, WtHR, FFMI, calcium, phosphorus, HDL-C, and glucose in relation to Vitamin D status. Furthermore, 25(OH)D levels were significantly higher in patients with Vitamin D sufficiency (*p* < 0.05).

Table 2 displays and compares the percentage values for the different cardiometabolic risk factors, analyzed according to Vitamin D status. The percentage of individuals who show blood pressure levels matching arterial hypertension (systolic or diastolic) is significantly higher in those individuals with hypovitaminosis (deficiency and insufficiency). In the same way, the percentage of individuals who present total-cholesterol values higher than 200 mg/dL, triglycerides values higher than 150 mg/dL, LDL-C values higher than 130 mg/dL, HDL-C values lower than 40 mg/dL, and HOMA-IR index values higher than 2.5 is significantly higher within the group of individuals with hypovitaminosis D.

The correlation between 25(OH)D levels and anthropometric and biochemical characteristics is shown in Table 3. There is a significant negative correlation (*p* < 0.05) of serum 25(OH)D levels with body fat percentage, FMI, systolic BP, total cholesterol, triglyceride, LDL-C, glucose, insulin, HOMA-IR, leptin, and PTH. In addition, there is a significant positive correlation (*p* < 0.005) of serum 25(OH)D levels and calcium and FFMI.

## 4. Discussion

This study demonstrates that hypovitaminosis D (insufficiency or deficiency) is quite a prevalent characteristic in those adolescents suffering from severe obesity, and, especially, that those subjects with Vitamin D deficiency (<20 ng/mL) have significantly high levels of different anthropometric (BMI z-score, waist circumference, total body fat, fat mass index, and systolic blood pressure) and biochemical (total cholesterol, LDL cholesterol, triglycerides, insulin, and HOMA-IR) measurements that imply cardiovascular risk in comparison to those individuals whose Vitamin D plasma levels are normal (≥30 ng/mL).

Blood sample analysis shows the following prevalence: Vitamin D sufficiency is present in 21.2% of the individuals, insufficiency in 39%, and deficiency in 39.8%, respectively. Needless to say, it might be thought that the coexistence of anthropometric and biochemical markers of cardiovascular risk in adolescents with severe obesity with a high prevalence of hypovitaminosis D is circumstantial, as a consequence of sedentary lifestyle and habits that lead to a progressive accumulation of fat mass, in addition to an alleged decreased outdoor activity. In fact, particularly obesity has been independently associated with low 25(OH)D levels and dyslipidemia [2,8]. Nevertheless, different authors have described—as it occurred in this study—the existence of an association between Vitamin D status and lipid profile in children and adolescents [15,25,26,27].

The negative correlation between 25(OH)D serum concentration and the different components of the lipid profile in these individuals (total cholesterol, LDL-cholesterol, and triglycerides) has aroused the analysis of the effects that a Vitamin D supplementation might exert on the lipid profile, since we could potentially get to modify an important cardiovascular risk factor. However, results from randomized clinical trials, including the evaluation of Vitamin D supplementation both in children and adults, have provided inconsistent results [14,28,29,30,31]. Another option would be that dyslipidemia itself influences Vitamin D levels and not vice versa, since statin use seems to improve both the lipid profile and the levels of Vitamin D simultaneously [32]. Nevertheless, a recent study has identified Vitamin D deficiency as an independent predictor factor for dyslipidemia in children with obesity [33].

In the present study, as several authors have described [21,27,34,35,36], we found a significant inverse association between serum 25(OH)D concentrations and serum insulin and HOMA-IR. Observational studies have detected that lower 25(OH)D levels are associated with a higher prevalence of impaired glucose tolerance or diabetes type 2 [37,38,39]. Vitamin D receptors are known to exist in pancreatic tissue, and calcium plays an essential role in B-cell insulin secretion, which implies that Vitamin D deficiency could increase the risk of impaired glucose metabolism. Obviously, additional studies are needed to determine whether treatment with Vitamin D can improve insulin resistance.

The diagnostic criteria of the metabolic syndrome proposed by the IDF have opted for a “lipid-centric” theory, with special attention to dyslipidemia and/or fat distribution [22]. However, insulin resistance has been considered as a determining pathophysiological factor of metabolic syndrome [40]. In fact, it has been postulated that Vitamin D deficiency would condition insulin resistance due to mechanisms not fully understood, and, as a consequence, the lipolytic activity would increase; this fact could explain the elevation of total cholesterol, LDL-cholesterol, and triglyceride plasma levels we observed in adolescents suffering from Vitamin D deficiency.

On the other hand, several observational studies have revealed an inverse relationship between 25(OH)D levels and BMI in children with obesity [8,27,41]. However, as we observed in this study, other authors have not found such an association [21,42]. We found an inverse correlation of serum 25(OH)D concentrations with different anthropometric and biochemical measurements that are specific of adiposity, such as body fat percentage, fat mass index, and leptin. Although BMI is useful to define obesity [17,43], it provides limited information since it does not allow us to discriminate between fat mass and fat-free mass [44,45]. In fact, several authors recommended the use of total body fat percentage or fat mass index (FMI) in contrast to BMI, in order to diagnose and monitor childhood obesity, owing to the higher sensibility to detect changes in body fat [46]. The low serum levels of 25(OH)D in patients with obesity could be attributed to decreased active outdoor life and sun exposure [8], but liquid chromatography/mass spectroscopy has shown a positive correlation between Vitamin D in adipose tissue and serum 25(OH)D [47]. This would indicate that adipose tissue would be a storage site for Vitamin D and explain, on one hand, the existing correlations between 25(OH)D plasma levels and body fat percentage and fat mass index and, on the other hand, the correlation found between plasma concentrations of leptin and 25(OH)D, since leptin would reflect the organic fat reserve [48].

In compliance with several studies [6,12,37,49], we found that 25(OH)D deficiency in adolescents with severe obesity is associated with high blood pressure (systolic or diastolic). Several mechanisms have been proposed on this relationship, such as the role of Vitamin D as a regulator of the renin–angiotensin system or a modulator of renin-gene expression [50,51]. Even so, Vitamin D receptors are present in vascular smooth muscle, which suggests that vascular smooth muscle is a target organ of Vitamin D [52]. Despite this finding, several randomized controlled trials have failed to confirm that Vitamin D has the effect of decreasing blood pressure [9,53].

In accordance with most authors, the mean values of PTH concentrations are increased in relation to children and adolescents with normal nutrition status [54,55,56]. In our study, we found a correlation between PTH and 25(OH)D levels, and this would be consistent with the physiological feedback mechanism of Vitamin D on parathyroid hormone secretion. However, according to some authors, this secondary elevation of PTH might increase lipogenesis and, consequently, foster fat storage [6,9,57]. In fact, physiologic elevation of PTH levels has been postulated as an independent predictor of obesity [54].

An important limitation of our study is the cross-sectional design. Therefore, our findings reflect an association, but exclude causal inference about the effects of low Vitamin D status on cardiovascular risk factors. In addition, data about dietary patterns, physical activity, and sun exposure were not incorporated into the study, and could result in hypovitaminosis D or dyslipidemia.

## 5. Conclusions

In summary, low Vitamin D levels in adolescents with severe obesity were significantly associated with some cardiometabolic risk factors, including body mass index, waist circumference, fat mass index, high blood pressure, impaired lipid profile, and insulin resistance. Prospective randomized controlled trials are justified to determine whether increased outdoor activities or dietary Vitamin D supplements that increase 25(OH)D levels could decrease cardiovascular risk among adolescents with severe obesity.

## Figures and Tables

**Table 1 children-07-00010-t001:** Anthropometric and biochemical characteristics according to Vitamin D status.

	Deficiency *n* = 94	Insufficiency *n* = 92	Sufficiency *N* = 50	*p*-Value *
Age (year)	13.4 ± 1.6	13.4 ± 1.6	13.4 ± 1.0	0.997
Weight z-score	4.3 ± 1.1	4.3 ± 0.9	4.2 ± 0.6	0.661
Height z-score	0.8 ± 0.7	0.7 ± 0.9	0.9 ± 0.8	0.362
BMI z-score	4.3 ± 1.1	4.4 ± 0.9	3.8 ± 0.6	0.001
Waist circumference	105.4 ± 7.5	108.6 ± 7.3	101.9 ± 7.7	0.001
Waist z-score	3.4 ± 0.9	3.6 ± 0.8	3.2 ± 0.9	0.005
WtHR	0.64 ± 0.06	0.63 ± 0.03	0.65 ± 0.05	0.072
Body fat (%)	38.1 ± 3.9	36.8 ± 3.6	36.2 ± 3.6	0.014
FMI (kg/m^2^)	12.9 ± 2.0	12.5 ± 2.1	11.8 ± 1.7	0.001
FFMI (kg/m^2^)	21.0 ± 2.7	21.8 ± 1.3	21.2 ± 2.1	0.401
Systolic BP (mm Hg)	132.3 ± 10.1	127.9 ± 11.5	125.2 ± 9.5	0.003
Diastolic BP (mm Hg)	76.8 ± 9.8	73.3 ± 9.3	73.8 ± 9.4	0.036
Calcium (mg/dL)	9.7 ± 0.3	9.8 ± 0.3	9.9 ± 0.2	0.431
Phosphorus (mg/dL)	4.3 ± 0.6	4.3 ± 0.5	4.2 ± 0.4	0.545
Total cholesterol (mg/dL)	165.5 ± 30.4	160.6 ± 31.4	147.2 ± 29.1	0.022
Triglycerides (mg/dL)	126.0 ± 39.5	95.8 ± 34.4	90.0 ± 30.2	0.001
HDL-C (mg/dL)	40.3 ± 7.4	43.8 ± 8.2	41.5 ± 8.2	0.113
LDL-C (mg/dL)	100.5 ± 24.6	96.5 ± 21.8	86.5 ± 25.6	0.034
Glucose (mg/dL)	89.3 ± 6.9	88.8 ± 8.1	86.6 ± 9.7	0.152
Insulin (uU/mL)	40.0 ± 23.9	26.6 ± 15.9	24.2 ± 10.1	0.003
HOMA-IR	9.2 ± 8.7	5.6 ± 3.9	5.4 ± 2.3	0.007
Leptin (ng/mL)	44.4 ± 13.3	38.7 ± 12.3	36.5 ± 13.9	0.026
Calcidiol (ng/mL)	13.9 ± 3.7	23.6 ± 2.6	36.5 ± 4.7	0.001
PTH (pg/mL)	58.9 ± 19.1	53.8 ± 15.1	44.3 ± 15.0	0.001

(*) ANOVA; BMI: body mass index. WtHR: waist-to-height ratio. FMI: fat mass index. FFMI: fat-free mass index. BP: Blood pressure. HDL-C: high-density lipoprotein cholesterol LDL-C: low-density lipoprotein cholesterol HOMA-IR: Homeostasis Model Assessment of Insulin Resistance. PTH: parathyroid hormone.

**Table 2 children-07-00010-t002:** Percentage of cardiometabolic risk factors analyzed according to Vitamin D status.

	Deficiency *n* (%)	Insufficiency *n* (%)	Sufficiency *n* (%)	*p*-Value *
Systolic BP				0.038
>130 mmHg	28 (37.8)	40 (55.6)	15 (37.5)	
<130 mmHg	46 (62.2)	32 (44.4)	25 (62.5)	
Diastolic BP				0.011
>85 mmHg	60 (81.1)	68 (94.4)	30 (75)	
>85 mmHg	14 (18.9)	4 (5.6)	10 (25)	
Total cholesterol				0.037
<200 mg/dL	76 (82.6)	84 (91.3)	45 (90)	
>200 mg/dL	16 (17.4)	8 (8.7)	5 (10)	
Triglycerides				0.002
<150 mg/dL	62 (70.5)	82 (91.1)	40 (80)	
>150 mg/dL	26 (29.5)	8 (8.9)	10 (20)	
HDL-C				0.028
>40 mg/dL	50 (56.8)	56 (62.2)	25 (50)	
<40 mg/dL	38 (43.1)	34 (37.8)	25 (50)	
LDL-C				0.053
<130 mg/dL	72 (35.8)	84 (41.8)	45 (22.4)	
>130 mg/dL	16 (59.3)	6 (22.2)	5 (18.5)	
Glucose				0.838
<100 mg/dL	86 (91.5)	80 (88.9)	45 (90)	
>100 mg/dL	8 (8.5)	10 (11.1)	5 (10)	
HOMA-IR				0.001
<2.5	22 (23.4)	44 (48.9)	15 (33.3)	
>2.5	72 (76.6)	46 (51.1)	30 (66.7)	

(*) χ^2^ test. BP: Blood pressure. HDL-C: high-density lipoprotein cholesterol LDL-C: low-density lipoprotein cholesterol HOMA-IR: Homeostasis Model Assessment of Insulin Resistance.

**Table 3 children-07-00010-t003:** Correlations of 25(OH)D with anthropometric and biochemical characteristics in bivariate analysis.

	Correlation Coefficient *	Significance
Weight z-score	−0.240	0.717
Height z-score	0.049	0.450
BMI z-score	0.057	0.381
Waist circumference	0.104	0.113
WC z-score	0.064	0.327
WtHR	0.052	0.423
Body fat (%)	−0.264	0.001
FMI (kg/m^2^)	−0.220	0.014
FFMI (kg/m^2^)	0.130	0.045
Systolic BP	−0.191	0.009
Diastolic BP	−0.067	0.363
Calcium	0.169	0.010
Phosphorus	−0.030	0.649
Total cholesterol	−0,268	0.001
Triglycerides	−0.270	0.001
LDL-C	−0.301	0.001
HDL-C	0.055	0.411
Glucose	−0.156	0.017
Insulin	−0.169	0.009
HOMA-IR	−0.160	0.015
Leptin	−0.191	0.022
PTH	−0.289	0.001

(*) Pearson’s correlation; BMI: body mass index. WC: waist circumference. WtHR: waist-to-height ratio. FMI: fat mass index. FFMI: fat-free mass index. BP: Blood pressure. HDL-C: high-density lipoprotein cholesterol LDL-C: low-density lipoprotein cholesterol HOMA-IR: Homeostasis Model Assessment of Insulin Resistance. PTH: parathyroid hormone.

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
