# Peer review of "Hypovitaminosis D and Cardiometabolic Risk Factors in Adolescents with Severe Obesity"

_children, 2020, doi:10.3390/children7020010_

Round 1
Reviewer 1 Report
Dear Authors, thank you very much for allowing me to review this paper.As you acknowledge in the beginning, the subject of Vit D and obesity is widely described in literature. Your paper adds an additional portion to this pool. General remark:1. Please bold p-values when statistically significant. It will make it more visible to the reader.2. In my opinion you can explain differences between the Vit D status groups without taking into account the fat%/BMI z-score of other obesity-related parameter into account. As you show in Table 1 - the groups significantly differ in the anthropometric measures. And this can be the main explanation of vit D status - as you also mention this in the introduction - vit D (especially 25OHD that was measured in blood) level is dependant on fat%.3. I would recommend you doing an additional analysis (regression) of all parameters (blood/anthropometric) depending on BMIz-score/fat%. This will give an additional info - yet still no full conclusion. 4. Additionally you suggest in lines 187-204 that all metabolic issues are solely associated with vit D - but you did not perform analysis with a dependent variable (bmi z-score/fat%) to exclude the impact of obesity on its own. This will probably change the result.5. In lines 212-218 you state that BMI is not associated with Vit D. As we can agree on your statement please take under the consideration that your groups did significantly differ in BMI z-scores dependent on the vit D level (Table 1)6. In lines 228-234 you once again state the vit D influences the BP - but you did not exclude the inpact of BMI/FMI/fat% on the result. Will it hold when you will adjust the result for bmi status?Specific remarks1. Did you check all the group for normality distribution of the data - I do not see that in your stat methods and yet you perform only parametric tests - were all the data normally distributed?2. Table 1. Height z-score - are you sure that the data are properly input in the table? Y Insufficiency group has height z-score 0.2 others 0.8/0.9 - and even with big SD there is no stat diff?3. Table 2 - the % should sum up in columns not in lines - as it is presented now it does not give any information at all. Will the results change? As I wrote earlier your paper is an interesting point of information from a country with presumably high sun levels showing still a high level of vit d deficiency. In my opinion after few additional tests it will be of a much greater value.
Author Response
We submit the corrections of the several aspects you sent us related to the manuscript entitled “Hypovitaminosis D and cardiometabolic risk factors in adolescents with severe obesity” (children-656649).
According to the suggestions of the reviewer, the following modifications have been made:
Reviewer 1
Dear Authors, thank you very much for allowing me to review this paper. As you acknowledge in the beginning, the subject of Vit D and obesity is widely described in literature. Your paper adds an additional portion to this pool. General remark:
1. Please bold p-values when statistically significant. It will make it more visible to the reader.
Statistically significant p-values ​​have been bolded. (tables 1, 2 and 3)
2. In my opinion you can explain differences between the Vit D status groups without taking into account the fat%/BMI z-score of other obesity-related parameter into account. As you show in Table 1 - the groups significantly differ in the anthropometric measures. And this can be the main explanation of vit D status - as you also mention this in the introduction - vit D (especially 25OHD that was measured in blood) level is dependant on fat%.
3. I would recommend you doing an additional analysis (regression) of all parameters (blood/anthropometric) depending on BMIz-score/fat%. This will give an additional info - yet still no full conclusion.
4. Additionally you suggest in lines 187-204 that all metabolic issues are solely associated with vit D - but you did not perform analysis with a dependent variable (bmi z-score/fat%) to exclude the impact of obesity on its own. This will probably change the result.
5. In lines 212-218 you state that BMI is not associated with Vit D. As we can agree on your statement please take under the consideration that your groups did significantly differ in BMI z-scores dependent on the vit D level (Table 1)
6. In lines 228-234 you once again state the vit D influences the BP - but you did not exclude the impact of BMI/FMI/fat% on the result. Will it hold when you will adjust the result for bmi status?
In this study, we have not found a relationship between 25(OH)D levels and BMI, but we found and inverse correlation of 25(OH)D with different anthropometric and biochemical measurements that are specific to adiposity (body fat percentage, fat mass index and leptin). Even though it is reasonable to think that the differences in anthropometric measurements could explain the different vitamin D levels/status, when achieving a multiple logistic regression analyses (dependent variable: BMI z-score, body fat percentage or FMI) with the totality of the participants group, as well as after splitting the sample into groups in relation with sex and pubertal stage, we found no significant results (independent factor associated).
We did not find any significant results (independent factor associated) either in the multiple logistic regression analyses when the dependent variable was vitamin D levels/status in relation to the metabolic factors analyzed. Likely, the fragmentation in subgroups might have conditioned the results obtained, and it might be necessary a larger sample in order to obtain more relevant results (independent factor associated).
We did not find any significant results in the regression analyses either (dependent variable: BMI/FMI/fody fat %) in relation with blood pressure.
The results of the multiple logistic regression analyses have not been included owing to the lack of significant results, but we remark the necessity to design prospective randomized controlled trials from the results obtained so as to determine whether increasing 25(OH) levels could help decrease cardiovascular risk in adolescents with severe obesity.
Specific remarks
1. Did you check all the group for normality distribution of the data - I do not see that in your stat methods and yet you perform only parametric tests - were all the data normally distributed?
Due to the sample size (all groups had more than 30 individuals) and considering the central limit theorem, the parametric tests have been used for the comparison of two or more group means (ANOVA).
2. Table 1. Height z-score - are you sure that the data are properly input in the table? Y Insufficiency group has height z-score 0.2 others 0.8/0.9 - and even with big SD there is no stat diff?
There is a transcription error in table 1. The value 0.2 ± 0.9 (height z-score) has been changed to 0.7 ± 0.9
3. Table 2 - the % should sum up in columns not in lines - as it is presented now it does not give any information at all. Will the results change?
The percentages previously arranged in line have now been placed in columns. It does not change the statistical significance.
As I wrote earlier your paper is an interesting point of information from a country with presumably high sun levels showing still a high level of vit d deficiency. In my opinion after few additional tests it will be of a much greater value.
We would like to express our thanks to the referee for your suggestions and positive criticisms.
We hope every made question have been answered adequately.
Yours sincerely,

Reviewer 2 Report
Congratulations to the study authors for a well-designed examination of metabolic and vitamin D status of the sample of adolescents with severe obesity in their region.
Specifically, the authors examined a cross-sectional sample of adolescents with severe obesity in this report [236 adolescents (154 boys and 82 girls),(BMI z-score >3.0, 99th percentile), aged 10.2 –15.8 years] and assessed the correlation of vitamin D status with cardiometabolic risk factors as well as calcium, phosphorus, PTH and vitamin D.
The authors note interesting findings, viz, a significant negative correlation of vitamin D levels with body fat percentage, Fat Mass Index, systolic BP, total cholesterol, triglyceride, LDL-C, glucose, insulin, HOMA-IR, leptin and PTH. These are in line with observations in other populations.
In line 195, authors note that other studies have shown vitamin D level/status to be in independent factor associated with dyslipidemia in addition to obesity. Therefore, regression analyses would be helpful to know whether this is relevant for the sample of the population studied as well since this has implications for designing interventional studies. Similarly, whether measures of insulin resistance are independent risk factors for vitamin D deficiency/insufficiency will provide more strength and relevance to the findings of the study.
Author Response
Congratulations to the study authors for a well-designed examination of metabolic and vitamin D status of the sample of adolescents with severe obesity in their region.
Specifically, the authors examined a cross-sectional sample of adolescents with severe obesity in this report [236 adolescents (154 boys and 82 girls),(BMI z-score >3.0, 99th percentile), aged 10.2 –15.8 years] and assessed the correlation of vitamin D status with cardiometabolic risk factors as well as calcium, phosphorus, PTH and vitamin D.
The authors note interesting findings, viz, a significant negative correlation of vitamin D levels with body fat percentage, Fat Mass Index, systolic BP, total cholesterol, triglyceride, LDL-C, glucose, insulin, HOMA-IR, leptin and PTH. These are in line with observations in other populations.
In line 195, authors note that other studies have shown vitamin D level/status to be in independent factor associated with dyslipidemia in addition to obesity. Therefore, regression analyses would be helpful to know whether this is relevant for the sample of the population studied as well since this has implications for designing interventional studies. Similarly, whether measures of insulin resistance are independent risk factors for vitamin D deficiency/insufficiency will provide more strength and relevance to the findings of the study
We totally agree with your considerations regarding the usefulness of the regression analyses.
In fact, we made the analyses with the totality of the group of participants, as well as after splitting the sample in relation with sex and pubertal stage and we did not find any significant associations between vitamin D level/status (dependent variable) and the cardiometabolic risk factors analyzed. Presumably, the fragmentations into subgroups might have conditioned the results we have obtained, and a larger sample size might be necessary in order to obtain relevant results (independent factor associated).
We would like to express our thanks to the referee for the positive criticisms.
Yours sincerely,

Reviewer 3 Report
This is a well written manuscript on the topic of vitamin D deficiency and cardiometabolic risk factors. This cross sectional study confirms findings previously published in the pediatric literature. My edit suggestions are as follows:
1. In the title and within the manuscript (i.e page 1, line 30 and line 40, etc), the phrase "cardiometabolic risks factors," is incorrectly stated. Please correct to "cardiometabolic risk factors" in the manuscript.
2. On page 2 lines 49-50, the authors state "...they would be the optimum population group in order to attempt to analyze a causal relationship between both entities." The authors are unable to analyze a causal relationship in a cross sectional study and the statement in the introduction may be misleading to the reader. The author should remove this statement and/or revise accordingly.
3. In the results section, page 3 line 130-133 states: "25(OH)D levels exceed 30 ng/mL (Vitamin D sufficiency) in 50 individuals (21.2%), oscillate
131 between 20 and 29 ng/mL (Vitamin D insufficiency) in 92 (39%) and are lower than 20 ng/mL (Vitamin D deficiency) in 94 (39.8%). This means, the prevalence of hypovitaminosis in adolescents with severe..."
However, there is no need to redefine vitamin D definitions as they were just defined in lines 112-115 on the same page. In addition, "exceed" should be changed to "exceeded" and the word "oscillate" is not appropriate in this context and should be changed. The phrase "This means" on line 132 should be removed.
4. On page 3, line 139, for the statement: "Obviously, 25(OH)D levels
were significantly higher in patients with vitamin D sufficiency (p<0,05)," the comma in the p value should be replaced with a period and the word "obviously" should be removed.
5. On page 4, line 152, HOMA-IR is incorrectly written as HOMA-RI and should be corrected.
6. On page 6, line 177, the authors should correct the vitamin D definition statement: "...normal (>30 ng/ml)" - normal was already previously defined in the manuscript as equal to or higher than 30ng/ml (page 3, line 115).
7. In the Discussion section, on page 6, in line 195 and line 213, the authors wrote "obese children." However, this is not people-first language and should be corrected to "children with obesity." This people-first language should be incorporated throughout the manuscript.
Author Response
We submit the corrections of the several aspects you sent us related to the manuscript entitled “Hypovitaminosis D and cardiometabolic risk factors in adolescents with severe obesity” (children-656649).
According to the suggestions of the reviewer, the following modifications have been made:
Reviewer 3
1. In the title and within the manuscript (i.e page 1, line 30 and line 40, etc), the phrase "cardiometabolic risks factors," is incorrectly stated. Please correct to "cardiometabolic risk factors" in the manuscript.
The word “risks…” has been changed to: “...risk…” (Title and lines 33, 40, 46 and 292 in the modified version).
2. On page 2 lines 49-50, the authors state "...they would be the optimum population group in order to attempt to analyze a causal relationship between both entities." The authors are unable to analyze a causal relationship in a cross sectional study and the statement in the introduction may be misleading to the reader. The author should remove this statement and/or revise accordingly.
Te word “causal…” has been removed (page 2, line 58 in the modified version).
3. In the results section, page 3 line 130-133 states: "25(OH)D levels exceed 30 ng/mL (Vitamin D sufficiency) in 50 individuals (21.2%), oscillate
131 between 20 and 29 ng/mL (Vitamin D insufficiency) in 92 (39%) and are lower than 20 ng/mL (Vitamin D deficiency) in 94 (39.8%). This means, the prevalence of hypovitaminosis in adolescents with severe..."
However, there is no need to redefine vitamin D definitions as they were just defined in lines 112-115 on the same page. In addition, "exceed" should be changed to "exceeded" and the word "oscillate" is not appropriate in this context and should be changed. The phrase "This means" on line 132 should be removed.
Where it previously said: “25(OH)D levels exceed 30 ng/mL (Vitamin D sufficiency) in 50 individuals (21.2%), oscillate between 20 and 29 ng/mL (Vitamin D insufficiency) in 92 (39%) and are lower than 20 ng/mL (Vitamin D deficiency) in 94 (39.8%). This means, the prevalence of hypovitaminosis in adolescents with severe..."
It is now expressed as: “...Ninety-four (39.8%) individuals had vitamin D deficiency, 92 (39%) vitamin D insuffiency and 50 (21.2%) vitamin D sufficiency. The prevalence of hypovitaminosis in adolescents with severe obesity was 78.8%“. (page 4, lines 153-155 in the modified version)
4. On page 3, line 139, for the statement: "Obviously, 25(OH)D levels were significantly higher in patients with vitamin D sufficiency (p<0,05)," the comma in the p value should be replaced with a period and the word "obviously" should be removed.
The word “Obviously…” has been removed. (page 5, line 163 in the modified version)
The comma in the p value (p<0,05) has been replaced with a period (p<0.05). (page 5, line 164 in the modified version)
5. On page 4, line 152, HOMA-IR is incorrectly written as HOMA-RI and should be corrected.
HOMA-RI has been changed to: HOMA-IR. (page 5, line 177 in the modified version)
6. On page 6, line 177, the authors should correct the vitamin D definition statement: "...normal (>30 ng/ml)" - normal was already previously defined in the manuscript as equal to or higher than 30ng/ml (page 3, line 115).
“…normal (>30 ng/ml)" has been changed to: “…normal (≥30 ng/ml). (page 7, line 207 in the modified version)
7. In the Discussion section, on page 6, in line 195 and line 213, the authors wrote "obese children." However, this is not people-first language and should be corrected to "children with obesity." This people-first language should be incorporated throughout the manuscript.
“Obese children” has been changed to “children with obesity” throughout the text. (page 8, lines 230 and 250 in the modified version)
We would like to express our thanks to the referee for the suggestions and positive criticisms that clearly have improved our manuscript.
We hope every made question have been answered adequately.
Yours sincerely,
